# Validation of SSUSI derived auroral electron densities: Comparisons to EISCAT data

Stefan Bender[1,2], Patrick J. Espy[1,2], and Larry J. Paxton[3]

[1]Department of Physics, Norwegian University of Science and Technology, Trondheim, Norway
[2]Birkeland Centre for Space Science, Bergen, Norway
[3]Applied Physics Laboratory, Johns Hopkins University, Laurel, Maryland, USA

**Correspondence:** Stefan Bender (stefan.bender@ntnu.no)

**Abstract.** The coupling of the atmosphere to the space environment has become recognized as an important driver of atmospheric chemistry and dynamics. In order to quantify the effects of particle precipitation on the atmosphere, reliable global energy inputs on spatial scales commensurate with particle precipitation variations are required. To that end, we have validated auroral electron densities derived from the Special Sensor Ultraviolet Spectrographic Imagers (SSUSI) data products for average electron energy and electron energy flux by comparing them to EISCAT electron density profiles. This comparison shows that SSUSI FUV observations can be used to provide ionization rate and electron density profiles throughout the auroral region. The SSUSI on board the Defense Meteorological Satellite Program (DMSP) Block 5D3 satellites provide nearly hourly, 3000-km wide high-resolution ($10\,\mathrm{km} \times 10\,\mathrm{km}$) UV snapshots of auroral emissions. These UV data have been converted to average energies and energy fluxes of precipitating electrons. Here we use those SSUSI-derived energies and fluxes as input to standard parametrizations in order to obtain ionization-rate and electron-density profiles in the E-region (90–150 km). These profiles are then compared to EISCAT ground-based electron density measurements. We compare the data from two satellites, DMSP F17 and F18, to the Tromsø UHF radar profiles. We find that differentiating between the magnetic local time (MLT) "morning" (03–11 h) and "evening" (15–23 h) provides the best fit to the ground-based data. The data agree well in the MLT "morning" sector using a Maxwellian electron spectrum, while in the "evening" sector using a Gaussian spectrum and accounting for back-scattered electrons achieved optimum agreement with EISCAT. Depending on the satellite and MLT period, the median of the differences varies between 0 and 20% above 105 km (F17) and $\pm 15\%$ above 100 km (F18). Because of the large density gradient below those altitudes, the relative differences get larger, albeit without a substantially increasing absolute difference, with virtually no statistically significant differences at the $1\sigma$ level.

## 1 Introduction

Particle precipitation and the processes initiated in the middle and upper atmosphere have been recognized as one ingredient to natural climate variability, and are included in the most recent climate prediction simulations initiated by the Intergovernmental Panel on Climate Change (IPCC) (Matthes et al., 2017). So far, however, most of the studies are based on in-situ particle observations at satellite orbital altitudes ($\approx 800\,\mathrm{km}$) (e.g. Wissing and Kallenrode, 2009; van de Kamp et al., 2016; Smith-Johnsen et al., 2018), or on trace-gas observations (Randall et al., 2009; Funke et al., 2017). In addition, most recent studies

focus on the influence of "medium-energy" electrons (30–1000 keV) (Smith-Johnsen et al., 2018) that have their largest impact in the mesosphere ($\lesssim 90$ km), but those occur more sporadically and have lower flux levels than typical lower-energy auroral electrons.

Here we present a method to estimate the auroral particle input from 90–150 km, which is not only larger than the medium-energy input, but also occurs more regularly and persists throughout the night. Subsequent chemical reactions result in auroral

particle precipitation being a major source of thermospheric NOx (Brasseur and Solomon, 2005), which can directly and indirectly influence the atmospheric ozone (Randall, 2005; Randall et al., 2009; Funke et al., 2005). To date, the impacts of this thermospheric source of aurorally produced reactive odd nitrogen (NOx) on the lower atmosphere are uncertain due to the insufficient altitude, spatial, and temporal sampling of currently used observations to characterize its source-function and transport to the stratosphere (e.g. Randall et al., 2001, 2009). Using direct auroral observations will help to elucidate and

quantify the production of auroral NOx with high spatio-temporal resolution, in particular as potential input for chemistry-climate models to trace the transport.

The Special Sensor Ultraviolet Spectrographic Imagers (SSUSI) are one of the "Special Sensor" instruments on each of the Defense Meteorological Satellite Program (DMSP) Block-5D3 satellites F17 and F18 (Paxton et al., 1992, 2017, 2018). These satellites orbit at 850 km altitude in polar, sun-synchronous orbits with equator crossing times of the ascending nodes of

40 17:34 (F17) and 20:00 (F18). The orbital period is of the order of 100 min such that an approximately 3000-km wide swath of the auroral zone is pictured multiple times by each satellite during a single night. The latest DMSP-5D3 satellites, F17–F19, were launched in 2006 (F17), 2009 (F18), and 2014 (F19). Here we compare the data from F17 and F18 to the ground-based measurements because control over F19 was lost in February 2016, and the observation time of F19 was apparently too short to facilitate a meaningful comparison.

The EISCAT (European Incoherent Scatter Scientific Association) incoherent scatter radars (ISR) are located in Northern Europe. They are located in Kiruna in Sweden, Sodankylä in Finland, Tromsø in Norway, and in Longyearbyen on Svalbard. Thus they are positioned approximately in the auroral zone at low and moderate geomagnetic activity, providing measurements of the ionospheric composition such as electron and ion densities and temperatures.

In a previous study, Aksnes et al. (2006) compared EISCAT radar data and UV-derived satellite data during a single day.

The satellite data were derived from the SSUSI predecessor sensors called UVI (Ultraviolet Imager; Torr et al. (1995)), and the study validated the optical approach, at least for moderate geomagnetic activity. In their study, Aksnes et al. (2006) compared the FUV-derived electron density profiles from 105 km to 155 km, with generally good agreement between UVI and EISCAT. They achieve that by individually choosing the precipitating electron spectrum for the UVI profiles that best reproduces the EISCAT profiles during that single substorm event.

A recent study by Knight et al. (2018) compared SSUSI-derived electron density parameters to ionosonde data. The altitude (hmE) and magnitude (NmE) of the ionospheric E-layer derived directly from the SSUSI UV observations were compared with the same parameters derived from the ionosondes. Their extensive analysis found also a good agreement between these parameters above 4 ionosonde stations at auroral latitudes longitudinally distributed around the globe. That study also contains an extensive review about the conversion of the SSUSI FUV data to the precipitating electron characteristics. In a follow-up

study, Knight (2021) also investigated the contribution of proton precipitation to the auroral emissions, finding a lower impact of protons than expected.

Here we use the SSUSI data for a full statistical investigation similar to the study presented by Knight et al. (2018), extending the earlier study by Aksnes et al. (2006) to multiple local times and auroral conditions. We also base our calculation on the approach presented in Aksnes et al. (2006), using the more recent ionization rate parametrizations introduced by Fang et al. (2010).

The manuscript is organized as follows, Sect. 2 introduces the SSUSI satellite data and the EISCAT radar data. In Sect. 3 we present the details of the comparison method, and in Sect. 4 we present our results and discuss them in Sect. 5. Our conclusions are then presented in Sect. 6.

## 2    Data

### 2.1    SSUSI UV and electron data

The SSUSI instruments remotely image the far-ultraviolet (FUV) auroral emissions (Paxton et al., 1992, 1993, 2002; Paxton and Zhang, 2016; Paxton et al., 2017). By scanning approximately $\pm 60°$ across track (Paxton et al., 1993), the SSUSI instruments observe the auroral zone on an approximately 3000 km wide swath. The single pixel resolution is $10 \times 10\,\mathrm{km}^2$ at the nadir point, and the scans extend from about $50°$ polewards in both hemispheres. The instantaneous field-of-view of the imaging spectrograph is $11.8°$, with 16 pixels along track, and overlapping across-track scans comprise the auroral swath as described in Paxton et al. (1992, 1993). The procedure also accounts for off-nadir effects in the FUV emissions (Paxton et al., 2017), and the processing steps are outlined in Paxton et al. (1993).

The SSUSI sensors record the FUV spectrum from 115 nm to 180 nm (Paxton et al., 1992), and they use in-flight calibration using a FUV star spectrum with well-understood brightness and spectral shape (Paxton et al., 2017). The downlink is limited to 5 channels with spectral centres at 121.6 nm (atomic hydrogen H Lyman-$\alpha$), 130.4 nm and 135.6 nm (both atomic oxygen OI), and 2 channels for the $N_2$ Lyman–Birge–Hopfield system (LBH), centred at 145 nm (140–150 nm, LBH-S) and 172.5 nm (165–180 nm, LBH-L). These channels capture the main auroral UV emissions, and are used to calculate the average electron energy, $\bar{E}$ in keV, and total electron energy flux, $Q_0$ in $\mathrm{erg\,cm}^{-2}\,\mathrm{s}^{-1}$, at each pixel (Strickland et al., 1983, 1999; Knight et al., 2018).

Here we use the data from the SSUSI sensors on board the DMSP F17 and F18 satellites over their respective operating periods from 2008–2019 and from 2011–2019. In particular, we use the SSUSI Level-2 "Auroral-EDR" (Environmental Data Record) data product for auroral electron energy and energy flux, which are derived from the $N_2$ LBH bands (Strickland et al., 1983). These quantities are provided in the environmental data records on a geomagnetic grid with a spacing of approximately 10 km $\times$ 10 km. The general algorithm for the SSUSI data is based on Strickland et al. (1999) and is described in Knight et al. (2018) and in detail in the SSUSI data product algorithms descriptions (available at https://ssusi.jhuapl.edu/data_algorithms, last access 17 June, 2021). To compare to EISCAT data, only data points within $2 \times 2°$ (latitude$\times$longitude) of the radar's

geomagnetic location were used. In addition, we require the average energy to be within the valid regime ($2\,\mathrm{keV} \leqslant \bar{E} \leqslant 20\,\mathrm{keV}$), and the derived energy flux $Q_0$ to be non-zero.

This corresponds to the range over which one can determine the characteristic energy of the precipitating electrons just based on the ratio of LBH long, assumed to have little or no $O_2$ absorption, to the LBH short which is assumed to be attenuated by $O_2$. "Soft" electrons, meaning low energy, dissipate their energy high in the atmosphere, and there is no $O_2$ absorption in the LBH short or long. This means the ratio is almost constant and determining the characteristic energy below about $2\,\mathrm{keV}$ becomes ambiguous. As the characteristic energy increases, the electrons are deposited deeper in the atmosphere. Eventually the $N_2$ LBH long emissions start to get attenuated, and deducing the flux from LBH long becomes ambiguous. This attenuation starts to become important at and below the deposition altitudes for $20\,\mathrm{keV}$ electrons, approximately $90\,\mathrm{km}$ Germany et al. (1990).

In addition, quenching losses of the $N_2$ LBH emissions constitute about 20% of the total deactivations at $90\,\mathrm{km}$, and cascade and collisional energy transfer begin to occur, which can also distort the spectral distribution of the LBH. While the modelling based on Strickland et al. (1999) includes both quenching and $O_2$ absorption, the complexity of the energy transfer in the singlet systems of the $N_2$ molecule (Ajello et al., 2020) made it prudent to limit the energy range retrieved from SSUSI to $20\,\mathrm{keV}$ to avoid large corrections. For $20\,\mathrm{keV}$ electrons, the LBH emission is falls extremely rapidly below $90\,\mathrm{km}$ (Germany et al., 1990).

## 2.2 EISCAT electron densities

We use data from the Tromsø UHF radar, which is located at 69°35'11"N and 19°13'38"E, in the auroral zone. The Tromsø radars include both transmitter and receiver, enabling them to provide altitude-resolved profiles of several ionospheric parameters, such as electron density, electron temperature, ion temperature, and many others, above the location using the incoherent scatter radar technique (Robinson and Vondrak, 1994; Lehtinen and Huuskonen, 1996). Depending on the so-called "pulse code" used for the "experiment", the altitude resolution can be less than $200\,\mathrm{m}$, but more typical in our comparison is $\approx 5\,\mathrm{km}$. In addition, the antennas of the Tromsø radars can be pointed in different directions and different altitudes.

We use the publicly available EISCAT E-region electron density data from the Tromsø UHF radar. The data are available via the "Madrigal" data base at http://cedar.openmadrigal.org (last access 21 September 2020). The data are averaged $\pm 5\,\mathrm{min}$ around the SSUSI scan time, and only high elevation angles $\geqslant 75°$ were considered. This time window was chosen so that several EISCAT profiles could be averaged to characterize the mean level of auroral ionization in the larger comparison region. No distinction between the different experiments (including scanning experiments) was made as long as there were electron densities available from at least $80\,\mathrm{km}$ and above, and all scans that provided those electron densities were interpolated onto a common 1-km altitude grid before averaging.

## 3 Method

There are a number of methods for treating atmospheric ionization from particle precipitation. These include multi-stream calculations (Basu et al., 1993; Strickland et al., 1993), derived parametrizations for spectra (Roble and Ridley, 1987; Fang

et al., 2008) and mono-energetic beams (Fang et al., 2010), and Monte-Carlo approaches (Schröter et al., 2006; Wissing and
Kallenrode, 2009). Similarly, numerous models are available for the recombination rates which are needed to calculate electron
densities from the electron–ion pairs produced by particle precipitation.

## 3.1 Ionization rates

We use the parametrization given by Fang et al. (2010) driven by the SSUSI-derived electron energies and fluxes, and combine
them with the NRLMSISE-00 (Picone et al., 2002) modelled neutral atmosphere to calculate the atmospheric ionization-rate
profiles. We use a Maxwellian spectrum for "morning" magnetic local times (MLT) (03–11 h) and a Gaussian for "evening"
MLT (15–23 h). Some care has to be taken when converting the average energy provided by SSUSI, $\bar{E}$, to the characteristic
energy $E_0$ required by those parametrizations. For the Maxwellian particle flux, the relation is $\bar{E} = 2E_0$, while for the Gaussian
the average energy is equal to the characteristic energy $\bar{E} = E_0$, and we set its width $W$ to $W = E_0/4$ (Strickland et al., 1983).
Before we use the parametrization by Fang et al. (2010), the total precipitating energy flux, $Q_0$, from the valid SSUSI data
points (those with non-zero $Q_0$ and $\bar{E}$ in the valid energy range as described in Sect. 2.1), are scaled by the ratio of the
number of valid observations to the total number of observations in the $2 \times 2°$ comparison area.[1] This is to compensate for the
portion of that area in which SSUSI did not observe sufficient UV emissions and thus could not infer the electron precipitation
characteristics properly.

The Fang et al. (2010) parametrization is derived for mono-energetic electron beams. We therefore integrate the ionization
rates $q_{\text{mono}}$ at altitude $h$ over the energy spectrum to obtain the total ionization rate $q(h)$ in $\text{cm}^{-3}\,\text{s}^{-1}$ at that altitude:

$$q(h) = \int_0^\infty q_{\text{mono}}(E, h)\phi(E)E\,\mathrm{d}E \ . \tag{1}$$

Here $\phi(E)$ is the electron differential flux in $\text{keV}^{-1}\,\text{cm}^{-2}\,\text{s}^{-1}$, the Maxwellian-type spectrum is given by (Fang et al., 2010,
Eq. (6)):

$$\phi(E) = \frac{Q_0}{2E_0^3} \cdot E \cdot \exp\{-E/E_0\} \ , \tag{2}$$

and the Gaussian particle flux spectrum[2] is given by (Strickland et al., 1993):

$$\phi(E) = \frac{Q_0}{\sqrt{\pi}W E_0} \cdot \exp\{-(E - E_0)^2/W^2\} \ . \tag{3}$$

In Eqs. (2) and (3), $E_0$ denotes the characteristic energy (mode of $\phi(E)$) in keV, and $Q_0$ is the total energy flux in $\text{keV}\,\text{cm}^{-2}\,\text{s}^{-1}$.

To convert energy dissipation into a number of electron–ion pairs, we similarly distinguish between early and late MLT.
This is due to the presence of upward moving back-scattered electrons contributing to the UV-derived flux (Rees, 1963; Banks

---

[1]Let $A$ be the set of all SSUSI points within the $2 \times 2°$ comparison area, and $B$ the set of valid points, i.e. the points used for the profile calculation defined
by $B := \{i \in A \mid 2\,\text{keV} \leqslant \bar{E}(i) \leqslant 20\,\text{keV} \wedge Q_0(i) > 0\}$. Then, the scaling we apply is equal to $Q_0(j) = \tilde{Q}_0(j) \cdot |B|/|A|$, $j \in B$; with $\tilde{Q}_0$ the flux given in
the SSUSI data files and $|\cdot|$ the cardinality of the sets.

[2]Note that the Gaussian distribution in Eq. (3) is normalized only when integrating from $-\infty \ldots \infty$. Integrating only the positive part leads to additional
terms of $\exp\{-E_0^2/W^2\}$ and $\text{erf}(-E_0/W)$ which can be neglected for sufficiently narrow distributions, i.e. large ratios of $E_0/W$.

et al., 1974; Basu et al., 1993; Strickland et al., 1993). This back-scattering effect depends on the type of auroral precipitation (Khazanov and Chen, 2021), and in our case seems to play a greater role at late MLT. We use the "standard" 35 eV per electron–ion pair (Porter et al., 1976; Roble and Ridley, 1987; Fang et al., 2008, 2010) for the early MLT, and to account for about 20% back-scattered electrons (Rees, 1963; Banks et al., 1974; Basu et al., 1993; Strickland et al., 1993), we use 43.73 eV per electron–ion pair for the late MLT.

In all the parametrizations used, the ionization rate $q$ is proportional to the ratio of the dissipated energy $\Delta E$ to the energy loss per electron–ion pair $\Delta\epsilon$, i.e. $q \propto \Delta E/\Delta\epsilon$. The dissipated energy $\Delta E$ is directly proportional to the incoming energy flux $Q_0$ and hence $\phi(E)$. Thus the aforementioned bounce effect can be accommodated either by reducing the effective energy flux (Basu et al., 1993; Strickland et al., 1993), or by increasing the energy required per ionization event. In this work we use 43.73 eV per electron–ion pair for the late MLT to effectively scale the energy flux as determined from the UV emissions.

## 3.2   Electron densities

Following Vondrak and Baron (1976); Gledhill (1986); Robinson and Vondrak (1994); Aksnes et al. (2006), the atmospheric electron density $n_e$ is related to the ionization rate $q$ by the recombination rate $\alpha$ via the continuity equation

$$\frac{\partial n_e}{\partial t} + \nabla \cdot (n_e \boldsymbol{v}) = q - \alpha n_e^2 \ . \tag{4}$$

Assuming a steady state and neglecting transport, for more details see, e.g., Vondrak and Baron (1976); Gledhill (1986);
Robinson and Vondrak (1994), we have $\partial n_e/\partial t = 0$ and $\boldsymbol{v} \approx 0$, which results in the relation $q = \alpha n_e^2$ or $n_e = \sqrt{q/\alpha}$.

   Different approaches have been used to parametrize the altitude dependence of the recombination rate $\alpha$ (Vondrak and Baron, 1976; Vickrey et al., 1982; Gledhill, 1986) and in the SSUSI data product algorithm descriptions (available at https://ssusi.jhuapl.edu/data_a last access 17 June, 2021). The simplest variant is a constant rate $\alpha = 3 \cdot 10^{-7}\,\mathrm{cm^3\,s^{-1}}$ (Vondrak and Baron, 1976), or an exponential relationship with a constant scale height of 51.2 km (Vickrey et al., 1982). Gledhill (1986) proposed the combination
of two exponentials with different scale heights for auroral inputs between 50 km and 150 km (Gledhill, 1986, Eq. (3)):

$$\alpha(h) = 4.3 \cdot 10^{-6} \exp\left\{-2.42 \cdot 10^{-2}\,h\right\} + 8.16 \cdot 10^{12} \exp\left\{-0.524\,h\right\}\,\mathrm{cm^3\,s^{-1}} \ . \tag{5}$$

This corresponds to scale heights of approximately 41 km at high altitudes and 2 km at the lower end. We use Eq. (5) as the better choice for the altitude range over which we compare the data, 90–150 km, and this is also consistent with Aksnes et al. (2006).

## 3.3   Comparison method

We follow the common approach for profile validation (e.g. Dupuy et al., 2009; Lossow et al., 2019), comparing the profiles of the absolute and relative differences together with their uncertainties (confidence intervals). For each orbit, the arithmetic mean $\mu_{\mathrm{orbit}}$ is calculated from all individual profiles derived from all valid SSUSI data points in the $2 \times 2°$ area around the radar (see Sect. 2.1 and footnote 1). For each corresponding orbit the average of the EISCAT electron densities within $\pm 5$ minutes of the

| MLT | F17 | F18 |
|---|---|---|
| 03–11 | 52 | 27 |
| 15–23 | 246 | 213 |

**Table 1.** Number of coincidences of F17 and F18 with the EISCAT Tromsø UHF radar during the two MLT sectors.

overpass, $\mu_{5\,\text{min}}$, is also calculated. The absolute difference of these quantities for each orbit at altitude $h$ is defined as:

$$\Delta N_{\text{e, orbit}}(h) = \mu_{\text{orbit}}(N_{\text{e,SSUSI}}(h)) - \mu_{5\,\text{min}}(N_{\text{e,EISCAT}}(h)) \, . \tag{6}$$

Thus positive values indicate larger electron densities from SSUSI and negative values imply larger EISCAT densities.

Here we compare the results of two remote-sensing instruments, each with their own uncertainties (see, e.g., Randall, 2003; Strong et al., 2008; Dupuy et al., 2009; Lossow et al., 2019). Thus we calculate the relative differences by dividing the absolute
differences by the average of the SSUSI and EISCAT densities:

$$\delta N_{\text{e, orbit}}(h) = \frac{2 \cdot \Delta N_{\text{e, orbit}}(h)}{\mu_{\text{orbit}}(N_{\text{e,SSUSI}}(h)) + \mu_{5\,\text{min}}(N_{\text{e,EISCAT}}(h))} \, . \tag{7}$$

We evaluate the distribution of those differences over all orbits by means of the 2.5th, 16th, 50th, 84th, and 97.5th percentiles. The 50th percentile is the median, the 16th and 84th percentiles correspond to the $1\sigma$, and the 2.5th and 97.5th percentiles to the $2\sigma$ confidence intervals. These percentiles are less susceptible to outliers and will give a better impression of the underlying
distribution than the mean and the standard deviation in cases where this distribution deviates substantially from a normal distribution.

## 4 Results

### 4.1 Available coincident data

An overview of the available coincident data between the SSUSI instruments and the Tromsø UHF radar is shown in Fig. 1. The
195 top panels within these figures show the distributions of the magnetic local times (MLT), which are for F17 centered around 05:40 h (downleg) and 19:20 h (upleg), and for F18 around 05:30 h (downleg) and 20:10 h (upleg), with a drift noticeable in both of the satellite orbits. The middle panels show the Kp values at the coincident overpasses, and the bottom panels show the radar elevation angles. The different colours represent different radar experiments (pulse codes) in which electron density profiles were collected.

The number of coincidences used in this study is summarized in Table 1. Note that there is an asymmetry between the data available for early and late MLT, with more coincidences during the latter. This imbalance, and possibly different precipitation characteristics during the different MLT, could lead to a possible bias in the calculated electron densities and their differences to the EISCAT measurements.

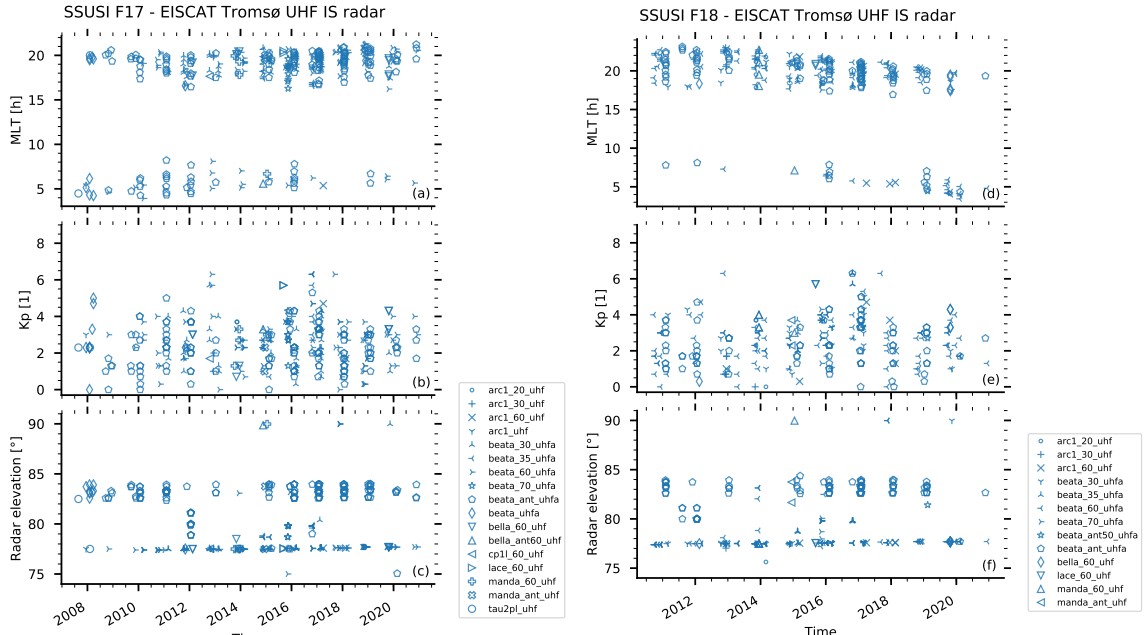

**Figure 1.** Available coincident data between the Tromsø UHF radar and the SSUSI on DMSP/F17 ((a)–(c)) and DMSP/F18 ((d)–(f)). Shown are the distributions of the data used in the comparisons according to their magentic local times (MLT, (a), (d)), the geomagnetic Kp index ((b), (e)), and the radar elevation angles ((c), (f)). The symbols indicate the different EISCAT experiments, the ones with "ant" indicate scanning experiments following the antenna. The MLT are divided according to the times given in the text.

## 4.2 Profile comparisons

As a measure of the distribution of the absolute and relative differences, we use the median together with the 68% ($\approx 1\sigma$) and 95% ($\approx 2\sigma$) confidence intervals derived from the 16th and 84th as well as the 2.5th and 97.5th percentiles, respectively. This enables us to quantify the differences better in cases where the distribution of those are skewed.

**MLT 03–11**

For early MLT (03–11 h), the electron density profiles together with the absolute and relative differences between the SSUSI-
210 derived electron densities and the EISCAT Tromsø UHF radar measurements are shown in Figs. 2 and 3. The profiles were calculated over all coincidences described in Sect. 3.3, using the "standard" parameters for the ionization rates as described in Sect. 3.1 and the "aurora" recombination rate parametrization from Gledhill (1986), see Eq. (3) therein or Eq. (5) above.

The F17 morning sector results show low absolute and relative differences that grow as one approaches the peak electron density. On the other hand, F18 shows a small and nearly constant absolute difference throughout the altitude range. In both
cases, the relative differences become large below the peak due to the decreasing mean electron density (the denominator in Eq. (7)).

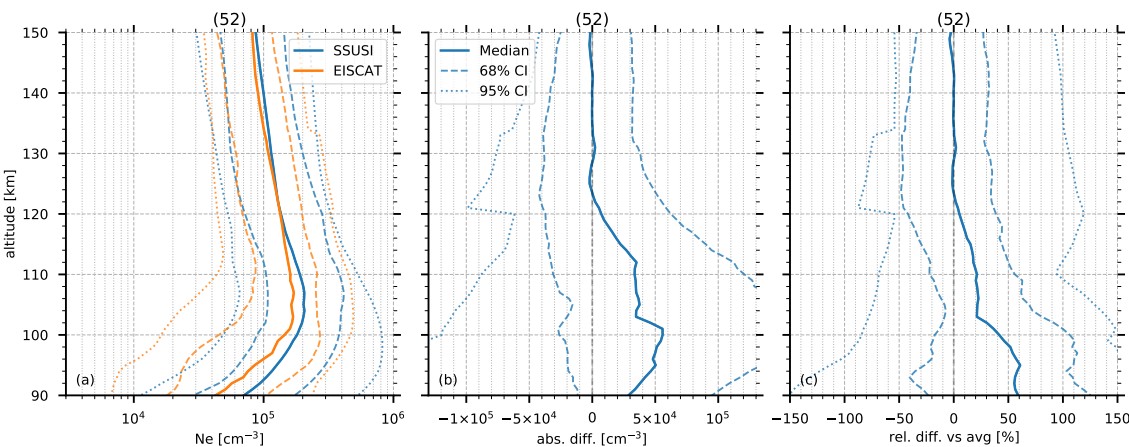

**Figure 2.** Profile comparison of calculated electron densities from SSUSI on DMSP/F17 to the ones measured by the EISCAT Tromsø UHF radar for early MLT (03–11 h). Density profiles (a), absolute differences (b), and relative differences (c). Shown are the medians (solid lines) and the 68% (dashed) and 95% (dotted) confidence intervals for the SSUSI-calculated electron densities (blue) and EISCAT (orange). The numbers in parentheses indicate the number of coincident satellite orbits used for averaging. The SSUSI profiles have been calculated assuming a Maxwellian electron spectrum with $E_0 = \bar{E}_{\mathrm{SSUSI}}/2$ and 35 eV / ion pair. Note that the density profiles (a) are on a logarithmic scale.

For F17 (Fig. 2), the median of the absolute differences grows from near zero above 120 km to about $6 \times 10^4$ cm$^{-3}$ (40%) at 100 km near the peak electron density. Below the peak, the absolute differences decrease to $3 \times 10^4$ cm$^{-3}$ near 90 km, but the relative differences increase due to the rapidly decreasing mean density. For F18 (Fig. 3), the median of the absolute differences remains between $-0.5$ and $+1 \times 10^4$ cm$^{-3}$ above the electron density peak near 100 km, leading to relative differences between $\pm 10\%$. Below the peak, absolute differences become $-1 \times 10^4$ cm$^{-3}$ at 90 km, and the magnitude of the relative differences again increases due to decreasing mean densities.

**MLT 15–23**

For late MLT (15–23 h), the electron density profiles and the absolute and relative differences between the SSUSI-derived electron densities and the EISCAT Tromsø UHF radar measurements are shown in Figs. 4 and 5. As for early MLT, the profiles were calculated over all coincidences, but using a Gaussian electron spectrum and slightly larger energy per ionization event as described in Sect. 3.1.

For the evening sector, both the SSUSI and EISCAT observations suggest a broader electron density peak than in the morning sector. Both F17 and F18 demonstrate small and nearly constant absolute differences with EISCAT over the entire altitude range. The dipole structure of the differences would indicate a systematically higher peak height for EISCAT relative to SSUSI, and once again, the relative differences grow below the peak due to the rapidly decreasing electron density.

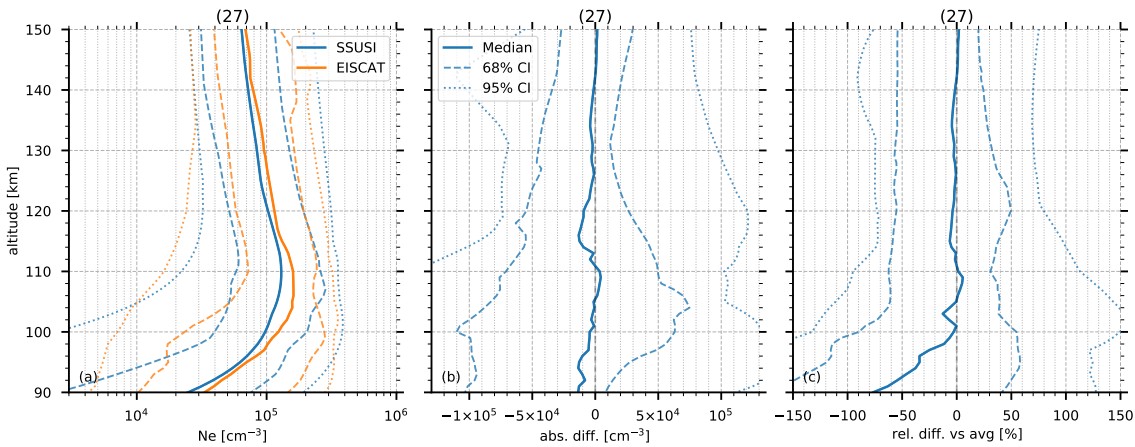

**Figure 3.** Profile comparison as in Fig. 2 for SSUSI on DMSP/F18 and the EISCAT Tromsø UHF radar for early MLT (03–11 h).

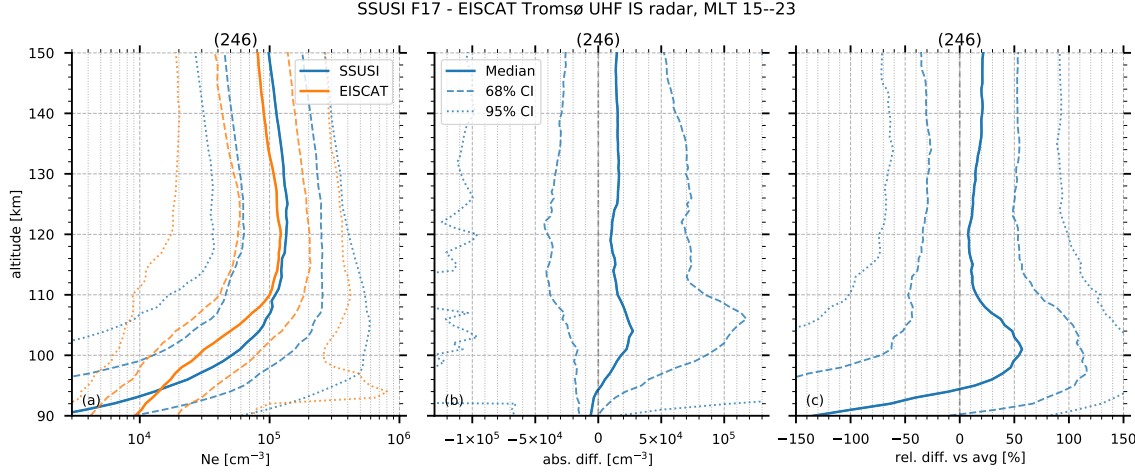

**Figure 4.** Profile comparison as in Fig. 2 for SSUSI on DMSP/F17 and the EISCAT Tromsø UHF radar for late MLT (15–23 h). The SSUSI profiles have been calculated assuming a Gaussian electron spectrum with $E_0 = \bar{E}_{\text{SSUSI}}$ and 43.73 eV / ion pair, details can be found in the text.

For F17 (Fig. 4), the median of the absolute differences is nearly constant at about $1 \times 10^4 \, \text{cm}^{-3}$ above 125 km (15–20%), and reaches $3 \times 10^4 \, \text{cm}^{-3}$ at 105 km (50%). While absolute differences decrease to about $0.5 \times 10^4 \, \text{cm}^{-3}$ at 90 km, relative differences again become large due to decreasing mean densities. For F18 (Fig. 5), both absolute and relative differences are nearly zero above 125 km. However, they reach $-1.5 \times 10^4 \, \text{cm}^{-3}$ (−15%) at 115 km, and $5 \times 10^3 \, \text{cm}^{-3}$ (10%) at 105 km. The absolute differences then decrease to $-1 \times 10^4 \, \text{cm}^{-3}$ at 90 km, again with large relative differences.

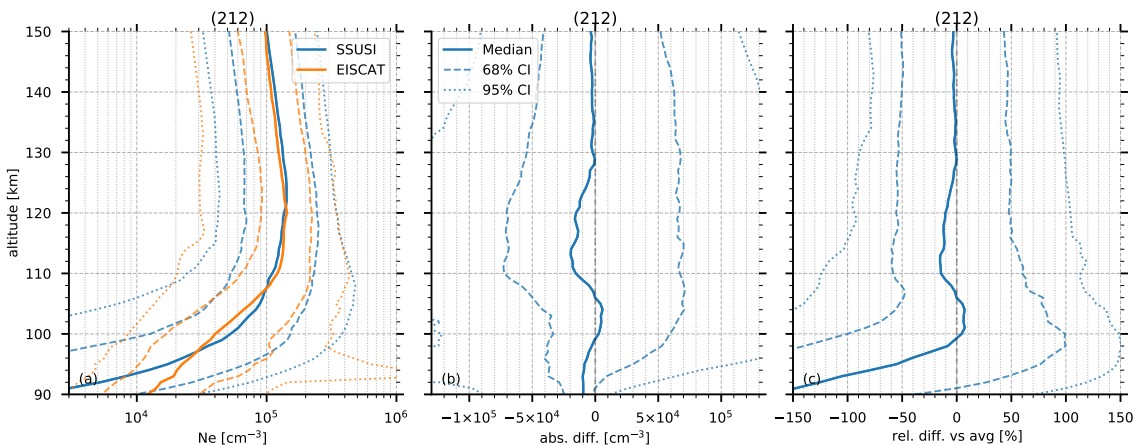

**Figure 5.** Profile comparison as in Fig. 4 for SSUSI on DMSP/F18 and the EISCAT Tromsø UHF radar for late MLT (15–23 h).

## 5   Discussion

In this study, we have used the mono-energetic approach derived by Fang et al. (2010) for atmospheric electron ionization rates, and integrated over Maxwellian and Gaussian particle spectra. Related parametrizations derived explicitly for Maxwellian particle flux spectra are available (Roble and Ridley, 1987; Fang et al., 2008), and the results for those are very close to the Maxwellian case studied here (not shown). Similarly, a variety of parametrizations exists for recombination rates, and here we chose the one given in Gledhill (1986). It should be noted that the parametrization by Vickrey et al. (1982) is very similar in the altitude region used in this study, resulting in comparable results.

The results show that the approach we have presented here, which mirrors an earlier study by Aksnes et al. (2006), leads to electron densities that agree with those measured by the ground-based EISCAT radars within the variability of the data. While more sophisticated approaches may lead to closer agreement between the different techniques, they are beyond the scope of this study. Such approaches would include calculating the ionization rates by solving a transport equation as in Basu et al. (1993), or using a fully relativistic approach (Wissing and Kallenrode, 2009). Both could help to improve the ionization rate profiles and the back-scatter ratio of the electrons, They would also enable different pitch-angle distributions to be used instead of relying on isotropic flux as in this study.

The differences observed between the different MLT sectors may be the result of different precipitation characteristics during the different times (Rees, 1969). Different magnetospheric acceleration mechanisms influence these characteristics, a short summary about the general mechanisms can be found, e.g., in Khazanov and Chen (2021). The observed emissions usually result from a mixture of these different auroral cases, and it seems that in our case, more back-scattering occurs during the evening MLT than during the morning MLT measurements. One should note that the "evening" MLT correspond to the beginning of the night and the auroral emissions just start to occur. In addition, the exact amount of back scattering has been

a debate for decades, ranging from 17% (Rees, 1963) to close to 50% (Banks et al., 1974), to presumably even higher values depending on the incident energy (Khazanov and Chen, 2021).

Note that the energy range provided by the SSUSI Auroral-EDR data is limited to 2–20 keV, which also limits the altitude range of comparable ionization rates to approximately 90–150 km (e.g Fang et al., 2008, 2010). The increasing (negative) differences between the SSUSI results and EISCAT at lower altitudes thus indicate the limits of the unambiguous energy range for the FUV-derived electron characteristics as described in Sect. 2.1.

It should be noted that the average energy and energy flux derived from the LBH emissions are essentially moments of the true distribution, such that one way to mitigate this problem may be assuming a different spectrum, for example adding a high-energy tail to the Maxwellian or Gaussian spectra (e.g. Strickland et al., 1993). However, the SSUSI energy range is typical for auroral inputs and good results at lower altitudes are not expected without further assumptions about the electron spectra. In addition, at lower altitudes the recombination rates increase substantially (Gledhill, 1986). This leads to increasing difficulties at lower altitudes when comparing observations of dynamic aurora by instruments with different observing volumes and spatio-temporal samplings as is the case here; the SSUSI instruments image a large area around the radar while the EISCAT is a narrow beam. Thus, future studies may employ ion-chemistry models such as the Sodankylä Ion Chemistry (SIC) model (Verronen et al., 2005; Turunen et al., 2009) to improve upon the recombination and quenching rates. Those models may also be used to derive trace gas species directly, which opens even more possibilities of comparisons, for example against satellite-based and ground-based trace gas measurements.

## 6   Conclusions

In this study we validate the electron density profiles derived from the SSUSI data products for effective energy and flux by comparing them to EISCAT derived electron density profiles. This comparison shows that SSUSI FUV observations can be used to provide high-resolution (down to $10 \times 10$ km) ionization rate profiles across its 3000 km wide swath within the auroral zone that are comparable to those measured by EISCAT between 100 and 150 km. In principle, the ionization rates can then also be used to calculate E-region conductivity and trace-gas profiles.

The data indicate that the comparison between the SSUSI volume measurements and the EISCAT narrow beam observations within that volume result in considerable pass-to-pass variability of the differences, caused by the wide range of auroral conditions and different precipitation characteristics. As a result, there are no statistically significant differences between the two measurement techniques. However, the trends in the comparisons show that a Maxwellian distribution and an energy loss per electron–ion pair of 35 eV is adequate for the morning sector (MLT 03–11). On the other hand, in the evening sector (MLT 15–23), where more back-scattered electrons are present, a Gaussian distribution with an energy loss of 43.73 eV per electron–ion pair is required to duplicate the higher and broader electron density peak.

The results show that electron densities derived from both SSUSI F17 and F18 agree with those measured by EISCAT to within 0–20% above 120 km. Although the differences are not statistically significant, the trend in the biases indicates that the SSUSI estimates are generally higher, and the differences are larger for the evening sector in comparison to the morning

sector. While SSUSI F18 maintains small, ≈10% differences with EISCAT through the peak of the electron density profile near 100 km, the trend of the SSUSI F17 bias tends to increase towards the peak, reaching as high as 40% before decreasing.

Below the peak density, the relative differences between EISCAT and both satellites become large due to the rapidly decreasing electron density. In addition, the SSUSI results tend to be smaller than the EISCAT densities below 95 km, indicating that the Maxwellian and Gaussian spectra may lack the high energies required to create ionization in this region. While the bias is
not significant, the tendency for SSUSI to underestimate the electron density at lower altitudes may be the result of the 20 keV limit of the SSUSI energy retrievals. This bias may also be due to the short recombination times in this region shortening the coherence times between the observations, and the parametrization failing to account for the formation of negative ions.

In virtually all cases (early and late MLT), the differences between EISCAT and SSUSI derived electron densities are well within the 68% ($\approx 1\sigma$) confidence interval derived from the distribution of the differences, and are always less than $2\sigma$. Thus,
the SSUSI instrument may be used to extend the EISCAT measurements across the auroral zone, quantifying both the auroral energy deposition and its spatial variability. Based on this work, future studies can further adjust the spectra as well as the recombination and quenching rates used for converting the UV emissions to electron energies and fluxes to match the ground-based measurements even better.

*Author contributions.* SB carried out the data analysis and set up the manuscript. PJE and LP contributed to the discussion and use of
language. All authors contributed to the interpretation and discussion of the method and the results as well as to the writing of the manuscript.

*Code and data availability.* The SSUSI data used in this study are available at `https://ssusi.jhuapl.edu/data_products` and the EISCAT data are available via the "Madrigal" database `http://cedar.openmadrigal.org`. The source code used to calculate the ionization rates and electron densities is available at `https://zenodo.org/record/4298137` (Bender, 2020) or upon request from the first author.

*Acknowledgements.* S.B. and P.J.E. acknowledge support from the Birkeland Center for Space Sciences (BCSS), supported by the Research Council of Norway under the grant number 223252/F50. L.P. is the principal investigator of the SSUSI project. EISCAT is an international association supported by research organisations in China (CRIRP), Finland (SA), Japan (NIPR and ISEE), Norway (NFR), Sweden (VR), and the United Kingdom (UKRI). The computations were performed on resources provided by UNINETT Sigma2 - the National Infrastructure for High Performance Computing and Data Storage in Norway. We further acknowledge the contributions by H. Knight to the discussion.

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
