# Peer review of "Validation of SSUSI derived auroral electron densities: Comparisons to EISCAT data"

_Annales Geophysicae, 2021_

## Author Response (AR1)

*We thank the reviewer for the comments that helped to improve the manuscript, we have answered the concerns and indicated the changes to the manuscript.*

The study describes a validation of SSUSI derived electron density profiles using ground-based electron density measurements by Eiscat incoherent scatter radar in Tromsø, Norway. The data were collected over 2 years and analyzed in 2 magnetic local time sectors separately: 03—11 & 15—23 MLT. Despite all averaging the agreement of the two data products is good within the analyzed height range of 90–150km. Thus, the results provide a promising outlook for spatially extending Eiscat electron density measurements with the help of SSUSI data.

**Comment:** The only bigger concern I have is the lack of description of the magnetic activity during the analyzed time periods. A brief look into a magnetic index data, as an additional parameter for Table 1, would provide some background on Eiscat location with respect to the auroral oval, i.e. what kind of precipitation the Eiscat radar was pointing to.

**Reply**: We thank the reviewer for this suggestion, which is a valid point. We are not sure that a single number such as the average Ap or Kp over all coincidences would be helpful, similarly for a list of geomagnetic index data for each single orbit. For SSUSI to observe sufficient UV emission to infer the average energy and flux, some geomagnetic activity is required, and to compare to EISCAT observations, the activity should not be too high because then the auroral oval would be south of the EISCAT radar. Thus the activity ranges from low to moderate Kp (0–4), with the most coincidences around a Kp of 2 for both MLT sectors.

**Action:**

We updated figure 1 to show the geomagnetic Kp index instead of the solar zenith angle in the middle panels. We also reduced the data shown therein to the data that were used in the comparison to reduce the complexity. The EISCAT experiments are now indicated by different symbols, and we added a legend, to also address one of the minor points the reviewer raised below.

Minor commentary:

- line 9: "derive" instead of "drive"?

**Reply:** We mean to "drive" as in using the data as input to the ionization and recombination parametrizations.

**Action:** To avoid confusion in this expression, we changed that phrase to: "...[energies and fluxes] as input to [standard parametrizations]..."

- line 25: more sporadic compared to what?

**Reply:** More sporadic than auroral electron precipitation with lower energies.

**Action:** We changed the sentence to read: "…, but those occur more sporadically and have lower flux levels than typical lower-energy auroral electrons."

- lines 27—30: so far only  particle precipitation and their energies have been talked about. Should this have an introductory sentences to say something about where NOx comes from?

**Reply:** We thank the reviewer for this suggestion, indeed there is a transition sentence missing. NOx is created by the reaction of the ionized or dissociated species with other ions or neutrals.

**Action:** We have added the following sentence: "Subsequent chemical reactions result in auroral particle precipitation being a major source of thermospheric NOx (Brasseur and Solomon, 2005), which can directly and indirectly influence the atmospheric ozone (Randall, 2005; Randall et al., 2009; Funke et al. 2005). To date, ..."

- lines 32-47: These few paragraphs are very detailed to be in the introduction. It would be more important to describe the Aksnes et al. results in more detail here and move some data and instrument details to the next section.

**Reply:** We thank the reviewers for this suggestion.

**Action:** We have shortened these introductory sentences to the essentials and moved the detailed description of the SSUSI data to Sect. 2.1, and of the EISCAT data to Sect. 2.2.

We added the following details about the Aksnes et al., 2006 study:

"In their study, Aksnes et al. (2006) compared the FUV-derived electron density profiles from 105 km to 155 km, with generally good agreement between UVI and EISCAT. They achieve that by individually choosing the precipitating electron spectrum for the UVI profiles that best reproduces the EISCAT profiles during that single substorm event."

- line 34: not sure if there is a point in giving time spans in both 0–12 and 0–24 time ranges..

**Reply:** They were intended to list the ascending and descending passes in the 00–24 time range, but we accidentally listed the UT times of the morning and evening coincidences with EISCAT.

**Action:** The times have been corrected to state the local equatorial crossing times of the ascending node of the satellites, 17:34 (F17) and 20:00 (F18).

- line 69: "at" seems misleading here

**Reply:** The reviewer is correct, its use is confusing in that sentence.

**Action:** We removed "at" from the sentence.

- line 70: I would not call experiments the same as pulse codes, as the latter is just one part of the experiment setup

**Reply:** We thank the reviewer for pointing that out, indeed they are not the same thing, and here we don't separate the pulse codes from the experiment.

**Action:** The sentences have been separated (see previous reply), and that part has been changed to: "In addition, the "experiment" run and pulse code used determine the altitude and time resolution."

We also changed "pulse codes" to "experiments" in (original) L73 to clarify that.

- line 72: what is the reasoning behind the analysis time window?

**Reply:** The 5 minute window was introduced to be long enough to have several EISCAT profiles to average, and short enough to avoid substantial changes in auroral activity.

**Action:** We added the following clarification:

"This time window was chosen so that several EISCAT profiles could be averaged to characterize the mean level of auroral ionization in the larger comparison region."

- line 74: were scanning experiments also included in this study?

**Reply:** Yes.

**Action:** We indicate which ones are scanning experiments in the caption of Fig. 1, and in the mentioned sentence in parentheses after "experiments": "experiments (including scanning experiments)..."

- footnote 1: the latter sentence could well be in the actual text

**Reply:** Thank you for the suggestion.

**Action:** We have included the whole footnote mentioned in the main text.

- section 3.1: The beginning of this section reads a little backwards. Following the data processing logic until the Fang et al. parametrization would probably be more logical..

**Reply:** We believe that introducing the bigger picture first, i.e. the parametrizations used, is the better approach. But we agree with the reviewer that the logical flow could be improved.

**Action:** We have swapped the order of the second and third sentence of that paragraph to put the description of the spectra before the details about the <E> to E0 conversion.

- line 120: what was the desire to be consistent with Aksnes et al. in the first place?

**Reply:** The consistency was not our main motivation, but it is a welcome side-effect.

**Action:** We changed that sentence to: "We use Eq. (5) as the better choice for the altitude range over which we compare the data, 90–150 km, and this is also consistent with Aksnes et al. (2006)."

- Figure 1: a legend to describe the experiments in the panels c & f would be helpful.

**Reply:** We thank the reviewer for this suggestion.

**Action:** We have updated the figure and included the legend as suggested. The 3 panels per satellite are now also consistent in that they use the same symbols for the different experiments.

- line 186: "is" instead of "are"

**Reply:** The phrase "a number of models" is used for denoting "several models" or "many models".

**Action:** To avoid confusion, we have changed the sentence to read: "..., numerous models are available ..."

- line 194: "study" instead of "studies"

**Reply:** Changed as suggested.

- line 200: "blindness of SSUSI" for the sake of completeness

**Reply:** We thank the reviewer for this suggestion, the term "blindness" was also misleading.

**Action:** We changed the sentence to: "...indicate the limits of the unambiguous energy range for the FUV-derived electron characteristics as described in Sect. 2.1."

- line 239: "short temporal and spatial scales" with respect to what? Although Eiscat provides data in high resolution it does not allow a high resolution extrapolation, so this is a confusing statement.

**Reply:** The reviewer is correct here, and the sentence did not convey the intended meaning.
**Action:** We have removed the phrase "short temporal and spatial scales" from the mentioned sentence.

**New references:**

Brasseur, G. P. and Solomon, S.: Aeronomy of the Middle Atmosphere, vol. 32 of Atmospheric and Oceanographic Sciences Library, Springer-Verlag, https://doi.org/10.1007/1-4020-3824-0, 2005.

Funke, B., López-Puertas, M., Gil-López, S., von Clarmann, T., Stiller, G. P., Fischer, H., and Kellmann, S.: Downward transport of upper atmospheric NOx into the polar stratosphere and lower mesosphere during the Antarctic 2003 and Arctic 2002/2003 winters, J. Geophys. Res. Atmos., 110, D24 308, https://doi.org/10.1029/2005JD006463, 2005.

Randall, C. E.: Stratospheric effects of energetic particle precipitation in 2003–2004, Geophys. Res. Lett., 32, L05 802, https://doi.org/10.1029/2004gl022003, 2005.

*We thank the reviewer for the comments that helped to improve the manuscript, we have answered the concerns and indicated the changes to the manuscript.*

This work compares E-region (90-150 km) electron density profiles, calculated from the average energy of auroral particles as inferred from ultraviolet images measured by two SSUSI instruments, to coincident ground-based electron density profiles measured by the EISCAT radar. The data are grouped by magnetic local time, with comparisons showing good agreement at magnetic local times 3-11 hr using a Maxwellian electron energy spectrum, and at 15-23 hr using a Gaussian spectrum with an increased value for energy needed for creation of each ion-electron pair to account for bounce-electrons that affect the measured UV flux.

Overall, the manuscript presents some interesting and relevant results.

**Comment:** My biggest issue is this: the title, abstract and conclusion all refer to the validation of SSUSI products, but what is presented is not such a validation. The reason lies mostly in Equation 7 that is used to generate the main results in Figures 2-4. Normally a validation effort will assume one established data set is ground truth (EISCAT in this instance) and complete a comparison of the new data set (SSUSI) to this truth. Equation 7 does not do that – it includes the "truth" and the "evaluated" data in both the numerator and denominator. I would expect the denominator to contain just the truth value to show a proper relative deviation from those data. If, for example, the SSUSI results are systematically high, then the delta will be large, but the relative difference calculated using Equation 7 will also increase the denominator and mask/reduce the full offset from the EISCAT result that appears in the numerator. To demonstrate this hypothetically, say that SSUSI = 2xEISCAT, then Equation 6 yields 1xEISCAT or 100% offset from the ground truth. But then Equation 7 yields 1xEISCAT/(3xEISCAT) = 1/3, or 33% "relative difference." If the denominator only contains EISCAT, then Equation 7 yields 1.0, equivalent to the 100% error from Equation 6.

**Reply:** We agree with the reviewer that using relative quantities alone, without any reference value, to quantify the agreement or disagreement of two instruments can be misleading. However, we presented the absolute difference profiles and relative difference profiles referenced to the average density profiles from both instruments. Using the average to base relative differences between two instruments on is common in data validation practice, in particular when comparing remote-sensing instruments, each with their own uncertainties, against each other as is the case here (see, e.g., Randall et al., 2003, Strong et al., 2008, Dupuy et al., 2009). We have not assigned either instrument as the "true electron density", instead seeking to demonstrate that the two instruments are commensurate and can be used to complement each other. Thus, we believe using the average is justified as it is common practice.

**Action:** We have extended the introduction of Eq. (7) and reiterate our reasoning to use the average, including more references:

"Here we compare the results of two remote-sensing instruments (see, e.g., Randall et al., 2003, Strong et al., 2008, Dupuy et al., 2009, Lossow et al., 2019), thus we calculate the relative differences by dividing the absolute differences by the average of ...".

We also changed the title to:

"Validation of SSUSI derived auroral electron densities: Comparisons to EISCAT data",

and the validation description in the abstract to:

"To that end, we have validated auroral electron densities derived from the Special Sensor Ultraviolet Spectrographic Imagers (SSUSI) data products for average electron energy and electron energy flux by comparing them to EISCAT electron density profiles."

**Comment:** Second, the manuscript needs to include a reference for the method of converting SSUSI spectra to auroral average energies, and provide a brief summary of the process to include which of the five SSUSI colors are used. This is important to address some areas of the discussion, for example, the valid energy range and associated altitude range under study and the level of increasing "blindness" to higher energies at lower altitudes. First mention is made on L64 of a "valid regime" that is not really explained until later in the manuscript at L198, and even then only briefly and without detail. Another example is later in the discussion (L232) that focuses on quenching of the airglow as a possible source of uncertainty at lower altitudes. While this seems a bit unlikely to me, more explanation of the process to derive energy from SSUSI data is needed to explain quenching as the driving factor of this. To that point, SSUSI only provides

an average energy value, not an altitude profile – I would be more suspicious of components in the analysis like the neutral density that has a direct impact on the altitude profile of ionization, especially since NRLMSIS-00 has been recently updated to MSIS 2.0 (Emmert et al., doi:10.1029/2020EA001321) and contains some key improvements at lower altitudes.

**Reply:** The data conversion is described by the algorithm description documents that accompany the SSUSI data, a peer-reviewed description can be found in (Knight et al., 2018). The electron energies and energy flux are derived from the two LBH bands. The scope of our study is to evaluate whether these data can be used to derive realistic ionization rate and electron density profiles in the E-region or not. We thank the reviewer for pointing us to the MSIS update. However, evaluating whether NRLMSISE-00 or MSIS 2.0 is the better choice for lower altitudes (we assume the reviewer is referring to < 95 km here) is beyond the scope of our study. In addition, MSIS 2.0 is patented and has an extremely restrictive license which makes it very difficult to use in the framework of open science, while NRLMSISE-00 is public domain software and easily accessible. Quenching is mentioned as one of the reasons for the increasing "blindness" at lower altitudes. While the altitude at which quenching losses are equal to radiative losses is about 80 km, they constitute about 20% of total deactivations at 90 km. While the Strickland modelling includes both quenching and $O_2$ absorption, the complexity of the energy transfer in the singlet systems of the N2 molecule made it prudent to limit the energy range retrieved from SSUSI to 20 keV to avoid large corrections. For 20 keV electrons, the LBH emission is seen to fall extremely rapidly below 90 km (doi.org/10.1029/JA095iA06p07725).

**Action:** We have updated the text to explicitly mention the data product we used for this study:

"In particular, we use the SSUSI Level-2 'Auroral-EDR' (Environmental Data Record) data product for auroral electron energy and energy flux, which are derived from the $N_2$ LBH bands (Strickland et al., 1983). These quantities are provided in the environmental data records on a geomagnetic grid with a spacing of approximately 10 km x 10 km. The general algorithm for the SSUSI data is based on (Strickland et al., 1999) and is described in (Knight et al., 2018) and in detail in the SSUSI data product algorithms descriptions (available at https://ssusi.jhuapl.edu/data_algorithms, last access 17 June, 2021).

We also added the following paragraph to Sect. 2.1 to briefly explain the limits of deriving the precipitating electron characteristics from the FUV emissions:

"This corresponds to the range over which one can determine the characteristic energy of the precipitating electrons just based on the ratio of LBH long, assumed to have little or no $O_2$ absorption, to the LBH short which is assumed to be attenuated by $O_2$. "Soft" electrons, meaning low energy, dissipate their energy high in the atmosphere, and there is no $O_2$ absorption in the LBH short or long. This means the ratio is almost constant and determining the characteristic energy below about 2 keV becomes ambiguous. As the characteristic energy increases, the electrons are deposited deeper in the atmosphere. Eventually the $N_2$ LBH long emissions start to get attenuated, and deducing the flux from LBH long becomes ambiguous. This attenuation starts to become important at and below the deposition altitudes for 20 keV electrons, approximately 90 km (Germany et al., 1990).

In addition, quenching losses of the $N_2$ LBH emissions constitute about 20% of the total deactivations at 90 km, and cascade and collisional energy transfer begin to occur, which can also distort the spectral distribution of the LBH. While the modelling based on Strickland et al. (1999) includes both quenching and $O_2$ absorption, the complexity of the energy transfer in the singlet systems of the $N_2$ molecule (Ajello et al., 2020) made it prudent to limit the energy range retrieved from SSUSI to 20 keV to avoid large corrections. For 20 keV electrons, the LBH emission falls extremely rapidly below 90 km (Germany et al., 1990)."

**Comment:** It is also important to report on the presence or absence of any cross-validation studies between F17 and F18. If these have not been validated internally as such, then it might be more appropriate to treat F17 and F18 as separate data sets rather than a single SSUSI data set for validation. This is specifically relevant to understand if the larger discrepancy with F17 at lower altitudes, near the peak of electron density (Figures 2 in particular) is significant since F18 does not show the same issue. It may be that it is appropriate to claim validation of the F18 data but not the F17 data.

**Reply:** We do not mix F17 and F18, the data from both satellite instruments are handled completely independently. In addition, the satellites are in different orbits, passing at different MLT, and such a cross-validation study would not be meaningful and is out of scope for this study.

**Action:** None.

**Comment:** L65: Explain the conditions under which zero-value $Q_0$ are found – whether a detection limit of the UV measurement or just times when no auroral emissions are present. It is not clear why it is appropriate to eliminate these data from the averaging process, rather than include them as a valid zero-density data point (as explained on L86). The data may otherwise be biased high if zero-value data are unnecessarily tossed out. L86-87 suggests this is just a detectability issue – but not a reason to toss the result and reduce the number of "valid observations."

**Reply:** These cases are described in the algorithm descriptions documents that accompany the SSUSI "auroral-EDR" data we use, the link has been provided with the update to the point discussed above. The SSUSI instrument can observe typical airglow levels (Ding et al., 2020). Thus, $Q_0=0$ represents cases either where no emission over nightglow levels is observable, or that a valid solution of flux with an average energy within 2 to 20 keV cannot be determined from the data (see data product algorithm descriptions).

These points are not tossed out as the reviewer's comment suggests, but included by scaling the flux appropriately as described in the paper. However, this scaling of the flux is not the same as including zero profiles in the final electron density average because of the non-linear relationship between the flux Q (and thus ionization rate q as described in Fang et al., 2010) and the electron density ne via Eq. (4).

**Action:** We included the statement about the unambiguous energy range in the data files in the data description as indicated in the reply to the previous point.

**Comment:** It should be clarified if any optimization was done, particularly in the selection of 43.73 eV/electron-ion pair (L105) used for the Gaussian distribution of the afternoon data. A brief explanation should be included to explain this, whether tests done to find the best fit, or if these are just as-run results.

**Reply:** We use this value to account for 20% back-scattered electrons that don't contribute to the ionization (Rees, 1963; Banks et al., 1974; Basu et al., 1993; Strickland et al., 1993).No fitting was done and the 20% were assumed for all late MLT coincidences.

**Action:** We have included the references in the introductory lines of that paragraph:

"This is due to the presence of upward moving back-scattered electrons contributing to the UV-derived flux (Rees, 1963; Banks et al., 1974; Basu et al., 1993; Strickland et al., 1993). This back-scattering effect depends on the type of auroral precipitation (Khazanov and Chen, 2021), and in our case seems to play a greater role at late MLT. We use the "standard" 35 eV per electron–ion pair (Porter et al., 1976; Roble and Ridley, 1987; Fang et al., 2008, 2010) for the early MLT, and to account for about 20% back-scattered electrons (Rees, 1963; Banks et al., 1974; Basu et al., 1993; Strickland et al., 1993), we use 43.73 eV per electron–ion pair for the late MLT."

We continue with the discussion about the flux and energy dissipation relationship as a new paragraph, which now ends:: "In this work  we use 43.73 eV per electron--ion pair for the late MLT to effectively scale the energy flux as determined from the UV emissions."

We also changed all occurrences of "bounce-electrons" to "back-scattered electrons" as it is more precise.

**Comment:** As far as organization of the text, there is content in the introduction that would be better placed in the description of the measurements, e.g. L32-42 is better suited in Section 2.1, and some or all of L43-47 in Section 2.2. I think more explanation is needed in Section 2.1, related to part of my major concerns above, to fully describe the data, including why the energy range has restrictions, what they are, and how they connect to the altitude range under study. L183-197 also contains information that should be included much earlier, in the description of the methods chosen for this study.

**Reply:** We thank both reviewers for this similar suggestion on the organizational structure.

**Action:** The lines have been moved to Sects. 2.1 and 2.2 as suggested, and they include the discussion as indicated in our replies above. We have moved L183-187 to Sect. 3 (Method) as an introductory paragraph, the information mentioned in the following lines (assuming it is not a typo) was already included in Sect. 3.1 and 3.2.

**Comment:** L39-43: More detail is important here, regarding the scan time, in-track imaging, and sampling overlap of adjacent scans. The pixel resolution on L40 is better stated as 10 km x 10 km, although this is only at the nadir, so a range should be included that corresponds to the range of data used in the analysis. No information is provided as to any restrictions placed on the angle from nadir for the SSUSI overlap to keep the 10 km dimension of the bin appropriate for the overlap determination. Other details should address how the data were processed to account for satellite motion, and co-adding of bins or remapping of detector pixel (as noted on L60) into larger image bins for the data product. It would be helpful for several areas of the discussion to include a figure showing a scan and overlap with EISCAT as well, to orient the reader. This would also help in explaining the calculation of the mean of the profiles in L125-126, and in L208 to demonstrate the relative size of imaging areas between the two measurement methods.

**Reply:** The data processing and description is available in the algorithm description documents and is discussed in (Knight et al., 2018). The instrument scanning method and processing steps are laid out in (Paxton et al., 1992, 1993), and the algorithm to account for off-nadir corrections is described in (Paxton et al., 2017). We use level 2 EDR data, as described in our reply to a previous point, which are presented on a 10 km x 10 km geomagnetic grid, all the processing steps are done by the data providers before we use them to calculate the electron density profiles. Thus the details mentioned are described within the references cited in the text. The purpose of this work is to demonstrate that the publicly available SUSSI data can be used to derive electron densities that are commensurate with ground based radar and not to define or improve the SSUSI processing algorithms.

**Action:** We added the following details to the opening paragraph of the SSUSI data description:

"The instantaneous field-of-view of the imaging spectrograph is 11.8°, with 16 pixels along track, and overlapping across-track scans comprise the auroral swath as described in Paxton et al. (1992), Paxton et al. (1993). The procedure also accounts for off-nadir effects in the FUV emissions (Paxton et al., 2017), and the processing steps are outlined in Paxton et al., (1993)."

**Comment:** The numerous footnotes seem unnecessary and distracting – it would be reasonable and appropriate for the material there to be placed into the body of the manuscript.

**Action:** Footnotes 1 and 2 have been included in the main text.

Other comments:

L7: Explain "high-resolution" in what sense (temporal, spatial, spectral), and compared to what.

**Reply:** We intended to refer to the high spatial coverage of a 3000-km wide swath per orbit, with a resolution of 10 x 10 km, compared to other in-situ auroral particle measurements that are usually used.

**Action:** This part of the sentence has been changed to: "3000-km wide high-resolution (10 km x 10 km) UV snapshots of auroral emissions."

L32: Clarify if SSUSI is one of five sensors on each DMSP, or if there are five DMSPs containing a SSUSI instrument.

**Reply:** [the SSUSI instruments] are one of five "Special Sensor" instruments, to avoid confusion, we changed the sentence as shown below.

**Action:** We replaced "five" by "the" in this sentence: "[the SSUSI instruments] are one of the "Special Sensor" instruments on each of the DMSP Block-5D3 satellites."

L33: Please provide a more accessible reference for the SSUSI sensors, rather than just an organizational promotional document (i.e. something with a DOI).

**Reply:** More references to the SSUSI instruments are already listed in the text below.

**Action:** We added references to (Paxton et al., 1992, 2017) in the mentioned sentence.

L34: The "observing times" noted here are confusing and should be clarified. I think maybe UT is not intended here, and perhaps should be LT instead. In any case, it is not clear why the observing time is restricted – unless perhaps these are the times where overflight of EISCAT is available. If this is just noting the local time of the ascending/descending nodes of the sun-synchronous orbit, then that should be clarified (including which node). However, those nodes are defined at the equator and not as critical to the polar observations presented here.

**Reply:** We thank the reviewer for pointing out this mistake. We accidentally listed the UT times of the EISCAT coincidences.

**Action:** We list the local equatorial crossing times of the ascending node, 17:34 (F17) and 20:00 (F18).

L35-36: The logic of this statement is perhaps a bit crossed. It is more correct to say that ONLY the data from F17 and F18 are compared because F19 was lost. Separately, if F19 was launched in 2014 and lost in February 2016, it would be important to know if data are available in that year of collection that could still be analyzed.

**Reply:** There is not much coincident data available from F19, and probably because of the small sample size or for some other reason that needs to be investigated separately, the agreement with EISCAT was not nearly as good as for the other 2 satellites.

**Action:** We changed that sentence to: "Here we compare the data from F17 and F18 to the ground-based measurements because control over F19 was lost in February 2016, and the observation time of F19 was apparently too short to facilitate a meaningful comparison."

L39: Here again, it is not clear if this is a 12 vs 24 hour notation, or an ascending/descending node time range, or something else.

**Reply:** This part of the manuscript has been included in Sect. 2.2, and the sentence referred to here has been removed.

L41: Clarify that this is the auroral zone above a specific region, e.g. above EISCAT, that is imaged multiple times, not the entire auroral zone that takes a full day to build imagery as the Earth rotates under the satellite.

**Reply:** The extent of a SSUSI "image" was described in the sentence just before the mentioned one, an approximately 3000 km wide swath of the auroral zone is imaged each orbit, and during multiple orbits a day.

**Action:** This part has been shortened to: "The orbital period is of the order of 100 min such that an approximately 3000-km wide swath of the auroral zone is pictured multiple times by each satellite during a single night."

L49: Provide a reference for UVI, if possible.

**Reply:** We thank the reviewer for this suggestion.

**Action:** We include a reference to (Torr et al., 1995) for UVI.

L54: Section 4 presents results, Section 5 contains the Discussion.

**Reply:** We thank the reviewer for this suggestion.

**Action:** We added the "Discussion" section as suggested.

L58: Footnote 1, clarify the stellar calibration – certainly, there is more than a single star that is used, but there are also shortcomings to the approach (e.g. point sources in a wider field of regard) that do not mitigate some of the uncertainty in the data. Such systematic errors are important and linked to my request to include a report on cross-validation of F17 and F18 data.

**Reply:** The SSUSI star-based in-flight calibration is described in detail in (Paxton et al., 2017). As stated above, cross-validating F17 and F18 data is out of the scope of our study since we do not combine the data from both satellites in any way.

**Action:** We added a reference to (Paxton et al., 2017) to the text.

L67: Rectify the references here with what is on L45.

**Reply:** We thank the reviewer for pointing out the mismatch in our references.

**Action:** This text has been merged with a part of the introduction as suggested by both reviewers, and the sentence mentioned has been removed.

L70: Modify "as well as…" to make consistent with the rest of the sentence, perhaps as "and can be configured for a number of experiments…"

**Reply:** We thank the reviewer for this suggestion.

**Action:** Also taking into account the comment from reviewer #1, the text has been merged with parts from the introduction. In the process we have split the sentence and removed that last part.

L99: Explain why electron bounce is important in the afternoon sector but not the morning MLT sector.

**Reply:** This is an area of active research, but the results so far indicate that the electron precipitation patterns in the two MLT sectors are different, discrete vs diffuse aurora, as well as visible aurora and pulsating aurora (see, e.g., Rees et al. 1969, Khazanov et al., 2021).

**Action:** In addition to the aforementioned changes about the 20% back-scatter, we have added the following paragraph to the discussion:

"The differences observed between the different MLT sectors may be the result of different precipitation characteristics during the different times (Rees, 1969). Different magnetospheric acceleration mechanisms influence these characteristics, a short summary about the general mechanisms can be found, e.g., in Khazanov and Chen (2021). The observed emissions usually result from a mixture of these different auroral cases, and it seems that in our case, more back-scattering occurs during the evening MLT than during the morning MLT measurements. One should note that the "evening" MLT correspond to the beginning of the night and the auroral emissions just start to occur. In addition, the exact amount of back scattering has been a debate for decades, ranging from 17% (Rees, 1963) to close to 50% (Banks et al., 1974), to presumably even higher values depending on the incident energy (Khazanov and Chen, 2021)."

L112: Justification should be given for making the assumption of steady-state for an auroral zone precipitating particle region, which seems highly dynamic, or quantify the level of uncertainty added.

**Reply:** We assume that the reviewer has overlooked the references that follow immediately after the expression "... steady state and neglecting transport (Vondrak and Baron, 1976; Gledhill, 1986; Robinson and Vondrak, 1994)" that go into the details of this assumption.

**Action:** To make the connection clearer, we added "... for more details see, e.g., Vondrak and Baron (1976); Gledhill (1986); Robinson and Vondrak (1994), we have $\partial n_e / \partial t = 0$ and $v \approx 0$, which results in ...".

L115: Remove or provide specifics on the "SSUSI internal document" referenced here.

**Reply:** We intended to refer to the algorithm description document, but this was not clear as pointed out by the reviewer.

**Action:** "(SSUSI internal document)" has been replaced by "the SSUSI data product algorithm descriptions (available at https://ssusi.jhuapl.edu/data_algorithms, last access 17 June, 2021).".

L142: Modify "centered for all satellites…" to reflect something that notes that the bimodal MLT distribution is related to the ascending vs descending nodes of the orbit (I assume), and perhaps has more data in the evening because the zero-flux restriction cuts out proportionally more morning data (again, I assume…). This is important to understand, as this "imbalance" is noted in L148. It seems unlikely to be related to the coincident overflight availability, but rather to the filtering of samples that is done.

**Reply:** This imbalance may simply be the effect of the different precipitation patterns during the different MLT sectors, resulting in less observable UV emissions in the "morning" MLT sector, as we discuss later in the text.

**Action:** We have changed the sentence to list both times as suggested:

"..., which are for F17 centered around 05:40 h (downleg) and 19:20 h (upleg), and for F18 around 05:30 h (downleg) and 20:10 h (upleg), ..."

Figure 1: Add the notation on what the colors represent in the caption. More detail should be given on these "pulse codes" and why they might be relevant to their interpretation, accuracy, etc.

**Reply:** We agree with the reviewer that the colours were missing their labels, the term "pulse codes" was actually misleading as the colours indicate the EISCAT experiments, of which the pulse code is just a part. We don't differentiate between the experiments as long as they provided electron densities within the comparison altitude region.

**Action:** Also considering the comment from reviewer #1, this figure has been updated to use symbols for the different EISCAT experiments and includes legends to list these experiments. We have also replaced "pulse-code" by "experiment" in most occurrences, as indicated in our reply to reviewer #1.

L158: Refer back to equation 3 here as well.

**Reply:** We thank the reviewer for this suggestion.

**Action:** We refer to Eq. (3) at the mentioned location.

L196: Some discussion should be given to at least some of the differences and improvements that could be made, and their potential impact, even if they are not explicitly evaluated for this work.

**Reply:** Some improvements were indicated in the following paragraph, using a photo-chemical model could improve upon the recombination and quenching rates. We thus added a brief discussion about some possible improvements by using more involved approaches to calculating the energy dissipation and ionization profiles.

**Action:** We added the following sentences to the paragraph mentioned in the comment:

"Such approaches would include calculating the ionization rates by solving a transport equation as in Basu et al. (1993), or using a fully relativistic approach (Wissing and Kallenrode, 2009). Both could help to improve the ionization rate profiles and the back-scatter ratio of the electrons, They would also enable different pitch-angle distributions to be used instead of relying on isotropic flux as in this study."

L214: It is not clear how high-resolution data are really validated given the extensive group averaging that is completed in this study. To do so would require a more extensive point-by-point analysis to claim validation in a single 10deg x 10deg image bin.

**Reply:** We agree that this expression is misleading, as we did not compare the SSUSI data products, but the derived electron densities, with a high-resolution compared to the extent of the auroral swath. However, we believe that the mentioned point-by-point analysis would not help here, as it would result in the same averaging process to derive the difference profiles. For each orbit the mean of the point-by-point differences is the same as the difference of the means of the point-by-point data. Averaging is inevitable in a study as this one, and we refer the reviewer to the references given in the manuscript, in particular (Randall et al., 2003, Strong et al., 2008, Dupuy et al., 2009) about validating profile data from different remote-sensing instruments. We take the point of view that there is no binary classification of valid/invalid data when it comes to using the derived ionization-rate or electron-density profiles.

**Action:** We changed the first sentence of that paragraph to:

"In this study we validate the electron density profiles derived from the SSUSI data products for effective energy and flux by comparing them to EISCAT derived electron density profiles."

L218-220: These first two sentences are a bit confusing, particularly whether "variability" refers to the measurements of EISCAT, derived from SSUSI, or the comparison between the two. As I read it, this is saying that the variability in the sample set exceeds the uncertainty of the measurements, which does not equate to being a validation.

**Reply:** We refer to "variability" as the variability of cases that are compared, as in different auroral conditions and different precipitation characteristics during each overpass.

**Action:** We changed the first sentence mentioned to end as follows:

"... pass-to-pass variability of the differences, caused by the wide range of auroral conditions and different precipitation characteristics."

L226-232: Six different mis-spellings of the SSUSI acronym within these seven lines of text.

**Reply:** We thank the reviewer for catching the typos.

**Action:** The misspellings have been corrected.

L236: Typo, "virtually"

**Reply:** We thank the reviewer for catching the typo.

**Action:** Corrected.

**New references:**

Ajello, J. M., Evans, J. S., Veibell, V., Malone, C. P., Holsclaw, G. M., Hoskins, A. C., Lee, R. A., McClintock, W. E., Aryal, S., Eastes, R. W., and Schneider, N.: The UV Spectrum of the Lyman-Birge-Hopfield Band System of N 2 Induced by Cascading from Electron Impact, J. Geophys. Res. Space Phys., 125, https://doi.org/10.1029/2019ja027546, 2020.

Banks, P. M., Chappell, C. R., and Nagy, A. F.: A new model for the interaction of auroral electrons with the atmosphere: Spectral degradation, backscatter, optical emission, and ionization, J. Geophys. Res., 79, 1459–1470, https://doi.org/10.1029/ja079i010p01459, 1974.

Germany, G. A., Torr, M. R., Richards, P. G., and Torr, D. G.: The dependence of modeled OI 1356 and N 2 Lyman Birge Hopfield auroral emissions on the neutral atmosphere, J. Geophys. Res. Space Phys., 95, 7725, https://doi.org/10.1029/ja095ia06p07725, 1990.

Khazanov, G. V. and Chen, M. W.: Why Atmospheric Backscatter Is Important in the Formation of Electron Precipitation in the Diffuse Aurora, J. Geophys. Res. Space Phys., 126, https://doi.org/10.1029/2021ja029211, 2021.

Knight, H. K.: Auroral ionospheric E region parameters obtained from satellite- based far-ultraviolet and ground-based ionosonde observations – effects of proton precipitation, Ann. Geophys., 39, 105–118, https://doi.org/10.5194/angeo-39-105-2021, https://angeo.copernicus.org/articles/39/105/2021/, 2021.

Knight, H. K., Galkin, I. A., Reinisch, B. W., and Zhang, Y.: Auroral Ionospheric E Region Parameters Obtained From Satellite-Based Far Ultraviolet and Ground-Based Ionosonde Observations: Data, Methods, and Comparisons, J. Geophys. Res. Space Phys., 123, 6065–6089, https://doi.org/10.1029/2017ja024822, https://agupubs.onlinelibrary.wiley.com/doi/full/10.1029/2017JA024822, 2018.

Randall, C. E.: Validation of POAM III ozone: Comparisons with ozonesonde and satellite data, J. Geophys. Res. Atmos., 108, https://doi.org/10.1029/2002jd002944, 2003.

Rees, M.: Auroral electrons, Space Sci. Rev., 10, https://doi.org/10.1007/bf00203621, 1969.

Rees, M. H.: Auroral ionization and excitation by incident energetic electrons, Planet. Space Sci., 11, 1209–1218, https://doi.org/10.1016/0032-0633(63)90252-6, 1963.

Strickland, D., Bishop, J., Evans, J., Majeed, T., Shen, P., Cox, R., Link, R., and Huffman, R.: Atmospheric Ultraviolet Radiance Integrated Code (AURIC): theory, software architecture, inputs, and selected results, Journal of Quantitative Spectroscopy and Radiative Transfer, 62, 689–742, https://doi.org/10.1016/s0022-4073(98)00098-3, 1999.

Strong, K., Wolff, M. A., Kerzenmacher, T. E., Walker, K. A., Bernath, P. F., Blumenstock, T., Boone, C., Catoire, V., Coffey, M., Mazière, M. D., Demoulin, P., Duchatelet, P., Dupuy, E., Hannigan, J., Höpfner, M., Glatthor, N., Griffith, D. W. T., Jin, J. J., Jones, N., Jucks, K., Kuellmann, H., Kuttippurath, J., Lambert, A., Mahieu, E., McConnell, J. C., Mellqvist, J., Mikuteit, S., Murtagh, D. P., Notholt, J., Piccolo, C., Raspollini, P., Ridolfi, M., Robert, C., Schneider, M., Schrems, O., Semeniuk, K., Senten, C., Stiller, G. P., Strandberg, A., Taylor, J., Tétard, C., Toohey, M., Urban, J.,

Warneke, T., and Wood, S.: Validation of ACE-FTS N 2 O measurements, Atmos. Chem. Phys., 8, 4759–4786, https://doi.org/10.5194/acp-8-4759-2008, 2008.

Torr, M. R., Torr, D. G., Zukic, M., Johnson, R. B., Ajello, J., Banks, P., Clark, K., Cole, K., Keffer, C., Parks, G., Tsurutani, B., and Spann, J.: A far ultraviolet imager for the International Solar-Terrestrial Physics Mission, Space Sci. Rev., 71, 329–383, https://doi.org/10.1007/bf00751335, 1995.